# Inflorescence Emergence and Flowering Response of Olive Cultivars Grown in Olive Reference Collection of Portugal (ORCP)

**DOI:** 10.3390/plants12112086

**Published:** 2023-05-24

**Authors:** Carla Inês, Maria C. Gomez-Jimenez, António M. Cordeiro

**Affiliations:** 1Instituto Nacional de Investigação Agrária e Veterinária, I.P. (INIAV), UEIS Biotecnologia e Recursos Genéticos, Estrada de Gil Vaz—Apartado 6, 7350-404 Elvas, Portugal; antonio.cordeiro@iniav.pt; 2Plant Physiology, Faculty of Science, University of Extremadura, 06006 Badajoz, Spain; mcgomez@unex.es

**Keywords:** phenology, olive cultivars, climate change, (endo)dormancy, ecodormancy, reproductive cycle, cold stress, *Olea europaea* L.

## Abstract

In olive trees, fluctuations in the onset of phenological stages have been reported due to weather conditions. The present study analyses the reproductive phenology of 17 olive cultivars grown in Elvas (Portugal) in 3 consecutive years (2012–2014). Through 2017–2022, the phenological observations continued with four cultivars. The phenological observations followed the BBCH scale. Over the course of the observations, the bud burst (stage 51) occurred gradually later; a few cultivars did not follow this trend in 2013. The flower cluster totally expanded phase (stage 55) was achieved gradually earlier, and the period between stages 51–55 was shortened, especially in 2014. Date of bud burst showed a negative correlation with minimum temperature (Tmin) of November–December, and, in ‘Arbequina’ and ‘Cobrançosa’, the interval stage 51–55 showed a negative correlation with both the Tmin of February and the Tmax of April, whereas in ‘Galega Vulgar’ and ‘Picual’ there was instead a positive correlation with the Tmin of March. These two seemed to be more responsive to early warm weather, whereas ‘Arbequina’ and ‘Cobrançosa’ were less sensitive. This investigation revealed that olive cultivars behaved differently under the same environmental conditions and, in some genotypes, the ecodormancy release may be linked to endogenous factors in a stronger way.

## 1. Introduction

Olive orchards are a key component of agricultural systems of the Mediterranean basin, and since the end of the last century there has been considerable expansion of olive production at low latitudes [1,2]. *Olea europaea* L. is an evergreen tree species and its vegetative structures cease their growth in autumn and undergo a winter rest period lasting until favorable temperature conditions return in late winter–early spring [3,4]. Olive reproductive structures are axillary buds of the leaves of the previous year’s shoots [5]. Axillary buds are developed during the growing season (spring–autumn) and they last in a dormant state until reproductive bud burst in the next spring [3,4]. In the Northern Hemisphere, inflorescence (and flower) external development takes place in early spring and the release of floral buds from dormancy occurs whenever trees have been exposed to a long enough period of chilling temperatures [6,7]. During chilling accumulation, reversal of the cold already experienced might occur due to warm temperatures [8]. After bud break, the time to flowering decreases as temperatures increase, mainly maximum temperature [9,10]. When flowering begins, high temperatures shorten the process [11] and the tree quickly goes through the beginning of flowering to the first petals fallen stage and the end of flowering.

Dormancy is a physiologic state that plants enter to protect their buds from winter hardiness. Environmentally induced dormancy onset and release coordinate growth cessation and resumption, but how chilling-dependent dormancy release and flowering are regulated remains unclear. In olives, winter chilling is associated with a hormonal change between endogenous gibberellins and growth inhibitors as abscissic acid [7,12]. To transit from dormancy to bud burst, trees must be exposed to a particular period of chilling temperatures and warm temperatures, referred to as chilling requirements and heat requirements. The former–chilling requirements for bud break–is an intrinsic part of tree physiology that varies within cultivars [2,8]. Winter dormancy is divided into two stages, endodormancy and ecodormancy [3,13,14]. Floral bud endodormancy is a state defined by physiological inhibition of flowering during the bud’s growing season formation. Fulfillment of chilling requirements marks the transition point from endodormancy to ecodormancy [3,8]. At this point, buds regain competency to respond to external environmental factors and they remain ecodormant under unfavorable growth conditions such as cold temperatures, but quickly progress toward bud burst when favorable conditions are present [3]. Plants that do not receive sufficient chilling fail to transition to ecodormancy, which leads to failing to flower or flowering erratically in the spring [8,11].

In recent decades, many studies on modeling olive phenology have been made [9,15,16]. A large number of them were calibrated for a single location [10,17,18,19,20] and as such may not be suitable for predicting under different environmental conditions, so efforts have been made to achieve models with a geographically wide applicability. De Melo-Abreu et al. [8] proposed a sequential model for different cultivars from long-term Spanish and Portuguese phenological records. It is a two processes in chain-type model that predicts the date when dormancy ends after reaching the chilling requirements and also predicts the full flowering date after the accumulation of warmer temperatures. This model [8] simulates the hourly increment of chilling that is generally gathered by commonly used models [21,22,23]. In the endodormancy release calculations, when the temperature exceeds 20.7 °C, a constant number of already accumulated chilling units are nullified, and the model does not accumulate chilling hours below 0 °C [8]. The 0 °C limit was based upon the work of Orlandi et al. [15] and the optimal temperature for chilling was considered to be 7.3 °C. Numerous studies have shown that several consecutive days of high temperatures produce a significant reduction in the number of inflorescences [2,8,24]. Using 10 cultivars and weather data for modelling validation procedures showed that increases of both daily maximum and minimum temperature of 2 °C and 3 °C would result in some cultivars that require more chilling being unable to flower [8]. Even a 1 °C mean temperature rise would lead to some years with delayed flowering dates [8]. Using open-top-chambers, it was observed that increasing temperatures 4 °C above the actual ambient temperature in Córdoba (southern Spain) may lead to an advance of the date of flowering in the ‘Picual’ cultivar, an increase of pistil abortion, and a reduction in fruit set, conditions with negative effects on yield [25]. A recent study aimed to develop procedures for identifying site-specific adaptation measures for Mediterranean olive orchards to reduce the negative effects of climate change [10]. Using an improved version of the AdaptaOlive model [26], it was reported that increases in temperature from 2 °C in some locations caused lack of chilling in the ‘Picual’ cultivar; thus, increases of 3 °C caused flowering failure in 29%, 68% and 74% of the years, in Cordoba, Jerez and Seville, respectively [10]. In addition, flowering failure in 100% of the years was projected when temperature increased by 5 °C in Seville and Jerez, and for increases of 6 °C in Cordoba, 7 °C in Jaen and 9 °C in Granada.

The high temperatures have raised significant questions about olive species, because of the implementation of olive orchards in low latitude regions and, more recently, the threat of the climate change associated with the increase in air temperature [27]. As a result, not much attention has been given to the lower limit for the minimum temperatures, beyond which floral induction in olives may somehow reverse. Optimal inflorescence production has been reported to occur when diurnal temperature varies between 2–15 °C [28], 2–4 °C to 15.5–19 °C [29], and 1.7–4.4 °C to 15.6–18.3 °C [30]. Increasing nighttime chilling by lowering temperatures to 2.2– −1.2 °C significantly reduced inflorescence production by 71 and 90%, respectively, in the ‘Arbequina’ cultivar [31]. Additionally, the time of initial development of first inflorescences was progressively delayed with increased chilling temperatures [31]. This could become a threat in the future, at least in the Mediterranean region of the Iberian Peninsula. In the province of Córdoba (southern Spain), an increase of around 1.5 °C was reported in spring temperatures, but between 1996–2012 the increase was less marked in the first 3 months of the year, and there was even a drop in January and March’s minimum temperatures at some study sites [32]. The sharpest increases have moved to late spring, during April [32]. In Tassaout, northeast of Marrakech, Morocco, during the period 1972–2019, the average of minimum and maximum temperatures increased by approximately 0.28 °C and 0.41 °C each decade, respectively [33]. An assessment of the impacts of climate change on the potential yield of olive trees and grapevines was recently performed across the Côa region (Portugal) [34]. Olive models’ results showed promising future improvements in the high-altitude areas of central Côa. However, this will not be the scenario for the majority of olive growing areas of the Mediterranean basin. To investigate the link between climate and olive species, 5 400 years of olive tree dynamics from the ancient city of Tyre, Lebanon, showed that optimal fruiting scales closely with temperature [35]. The optimal annual average temperature for olive flowering (16.9 ± 0.3 °C) has existed at least since the Neolithic period [35]. According to the study’s projections, during the second half of the twenty-first century, temperature increases in Lebanon will have detrimental consequences on olive tree growth and olive oil production, especially in the country’s southern regions, which will become too hot for optimal flowering and fruiting.

In the present study, the phenological growth stages of inflorescence bud development of 17 olive cultivars grown in Elvas were examined during 3 consecutive years to evaluate species and varietal local response to major weather-related parameters, especially temperature. To further this exploration, four cultivars were selected (‘Arbequina’, ‘Picual’, ‘Galega Vulgar’ and ‘Cobrançosa’) and their phenological development was followed over a larger period (6 consecutive years). Statistical analyses were performed between phenological stages dates and maximum and minimum temperatures (Tmax and Tmin, respectively), from the previous autumn to the current inflorescence growing. Identifying significant relations within these parameters and cultivars highlight how resilient they could be under a changing climate.

## 2. Results

### 2.1. Climate Analysis

Climate characterization of the reference period (RP: 1983–2014) of the Elvas region showed that during the last 3 months of the year (October–November–December), monthly rainfall is between 75 mm and 85 mm (Figure 1). Throughout the first 5 months of the year (January–February–March–April–May), a new growing cycle (n), monthly total rainfall decreases to amounts between 40 mm and 60 mm. During the studied seasons October 2011–May 2014 (Flw 2012–Flw 2014), rainfall was highly irregular: months with very low and no rainfall included February 2012 (Figure 1); conversely, months with abnormally high precipitation were also recorded. Compared to the RP, in February 2014, the total recorded rainfall was more than 2-fold higher, and in March 2013 it was approximately 4-fold higher (Figure 1). This means a precipitation augmentation of 64.18 mm and 124.54 mm, respectively (Table 1). Another feature of this analysis was the occurrence of dry periods during a time in which they are not expected (autumn, winter and spring months).

Both the Tmax and Tmin of the local RP showed a gradual decrease during October–January, followed by an increasing behavior until May. January’s Tmin and Tmax initially varied between 5 °C and 15 °C, and in May approached 10 °C and 25 °C, respectively (Figure 1). During these three growing seasons (Flw 2012, Flw 2013 and Flw 2014), some significant increases and decreases of these monthly weather variables were observed. 

At the beginning of 2012, a drop in Tmin occurred (Figure 1). January and February’s Tmins were 2.28 °C and 4.55 °C below the RP average, respectively (Table 1). January of 2013 showed a Tmin higher than the RP, but during February a significant drop of 1.58 °C was observed (Table 1). Additionally, as a result of the year’s exceptionally high rainfall from January until late March, the average Tmax was always around 15 °C (Figure 1). Indeed, the Tmax average of March 2013 was 3.11 °C under the RP’s average, and this was a significant decrease of this parameter (Table 1). In 2014, January showed Tmin and Tmax values of 2.18 °C and 1.09 °C above the RP’s average for that month (Table 1). May’s Tmax averages in 2012 and 2014 were significantly higher, with 2.32 °C and 2.25 °C more, respectively, than the RP’s average (Table 1).

The autumn of 2013 (Flw 2014) was characterized by a pronounced decrease of the Tmin for November and December (Figure 1). During this period, even negative values were recorded, and compared to the RP, November’s Tmin was 2.86 °C lower and December’s 3.07 °C (Table 1).

### 2.2. Phenological Response of a Wide Group of Cultivars

The phenological response of 17 olive cultivars during 3 consecutive years (2012, 2013 and 2014) is shown in Figure 2. In these 3 seasons, variations were observed both among cultivars and through the years. In 2012, stage 51 (bud burst) was dominant on the tree canopy (X–51–X) for the first 35% of the cultivars on DOY 53, and the last cultivars reached this stage on DOY 60 (Table 2). In 2013, the first 35% of the cultivars showed stage 51 as dominant on the canopy on DOY 44, and the remaining cultivars gradually achieved this phenophase by DOY 65 (Table 2). In 2014, stage 51 started to be dominant on DOY 62 for 12% of the cultivars, and the majority of them (71%) showed this phenophase on DOY 69.

The phenological response until stage 55 (X–55–X) showed some kind of adjustments between genotypes. In 2012, 18% of the cultivars showed stage 55 on DOY 117, and by DOY 129 the majority of them had reached this phenophase (Figure 2). The interval between stages 51–55 took a minimum of 57 days and a maximum of 76 days (average 68 days) (Table 2). In 2013, 18% of the cultivars showed stage 55 on DOY 109, and the last genotypes (12% of total) showed it on DOY 120. The interval between stages 51–55 took a minimum of 49 days and a maximum of 70 days (average 61 days) (Table 2). In 2014, the first cultivars (18% of total) reaching stage 55 were observed on DOY 104, and the last ones were observed on DOY 118. Pre-flowering amplitude (stage 51–55) had a minimum duration of 34 days and a maximum of 51 days (average 40 days) (Table 2).

The beginning of flowering (stages 60/61) in 2013 occurred from DOY 131 to DOY 140; in 9 days, all cultivars reached this phenophase (Figure 2). The interval between stages 55–60/61 had a minimum duration of 13 days and a maximum of 26 days (average 20 days) (Table 2). In 2014, the stages 60/61 became dominant on trees’ canopy from DOY 122 to DOY 125; all studied cultivars achieved the beginning of flowering in 3 days (Figure 2). The time between stages 55–60/61 varied from 7 days to 19 days (average 15 days) (Table 2).

Flowering, in 2013, became the dominant stage (X–65–X) on trees’ canopy from DOY 137 to DOY 143 (2nd half of May) (Figure 2). In 6 days, all cultivars were full flowering. The amplitude between stages 60/61–65 had a minimum duration of 2 days and a maximum of 9 days (average 6 days) (Table 2). In 2014, the flowering phase was achieved by all cultivars from DOY 125 to DOY 128 (1st half of May) (Figure 2). In 3 days, all genotypes were fully flowering and the amplitudes between stages 60/61–65 were very short, between 2 and 4 days (average 3 days) (Table 2).

A faster inflorescence growth was observed even during the first phases, stages 51–53/54 (Figure 2). Indeed, the average phenological behavior followed a sigmoidal curve in 2012 and somewhat in 2013, but in 2014 it was more like a double sigmoidal curve, a consequence of the late stage 51 manifestation compared to the 2 previous years (Table 2). The delayed occurrence of stage 51 in 2014 led to a double sigmoidal curve as the phenological response had to be faster during the first stages (51–53) and slower in the stages 55/57 (Figure 2).

### 2.3. Four Cultivars Case Study

In this group of 17 olive tree cultivars are two of the most studied in the entire world (‘Arbequina’ and ‘Picual’), and the two most important Portuguese autochthonous olive tree cultivars (‘Galega Vulgar’ and ‘Cobrançosa’). The manifestation of stage 51 showed variations within the same cultivar and through the three seasons 2012–2014, but a strong diminution of the amplitude was noticeable between stages 51–55 in 2014 (Table 2). The biggest impact occurred in ‘Galega Vulgar’, with a 50% decrease of the stage 51–55 length (68 days in 2012; 34 days in 2014) (Table 2). ‘Cobrançosa’, ‘Arbequina’ and ‘Picual’ showed reductions of 31%, 35% and 36%, respectively (Table 2).

In order to observe how the dates of occurrence of stages 51 and 55, as well as the period between these two stages, can experience variations, box plot diagrams were created with data of 9 years (2012–2014 and 2017–2022). During the study experiment, until 2018, these four cultivars experienced stage 51 gradually later than the previous year (Figure 3c). On the other hand, in general until 2020, stage 55 occurred earlier than the previous year, with 2018 as the biggest exception (Figure 3a).

In general, through 2012–2022, this group of four cultivars experienced a decrease of the duration of the interval between stages 51–55, with the shortest occurring in 2021 (Figure 4a). It was observed that ‘Galega Vulgar’ and ‘Arbequina’ had a distribution of the data regarding time between stages 51 and 55 more symmetric than the other two cultivars (Figure 4b), and ‘Arbequina’ was consistently the cultivar with the shortest period between stages 51–55.

The correlation analysis between reproductive phenology and weather variables exposed more traits. For all of these four cultivars, a statistically meaningful, negative and strong (very strong in ‘Arbequina’) correlation was found between the occurrence of stage 51 and the previous year’s Tmin of November and December (Table 3). In ‘Galega Vulgar’ and ‘Picual’, a statistically meaningful, positive and very strong and strong correlation, respectively, was also found between the occurrence of stage 51 and the previous year’s Tmax of November (Table 3).

As for the occurrence of stage 55, a statistically meaningful, negative and strong (very strong in ‘Picual’) correlation was found between this stage and the year’s Tmin of February (Table 3). The correlation between the duration of the inflorescence growth between stages 51 and 55 and weather variables resulted in a more heterogeneous behavior within these cultivars. In ‘Arbequina’ and ‘Cobrançosa’, a statistically meaningful, negative and at least strong correlation was found between the length of the period 51–55 and both the year’s Tmin of February and Tmax of April (Table 3). In ‘Picual’, a statistically meaningful, positive and strong correlation was found between the length of the period 51–55 and the year’s Tmin of March, and a negative and strong correlation with the year’s Tmax of April (Table 3). In ‘Galega Vulgar’, a statistically meaningful, positive and strong correlation was found only between the length of the period 51–55 and the year’s Tmin of March (Table 3).

Additionally, a linear regression was performed between the date of occurrence of stage 51 and the number of days during the previous months of October, November and December with a Tmin equal to or less than 2.2 °C. The 2.2 °C limit was based upon the work of Malik and Bradford [23] in the ‘Arbequina’ cultivar. The parameter R^2^ ranged between 58.8% in ‘Arbequina’ and 71.1% in ‘Galega Vulgar’ (Figure 5). According to the linear equations (y = b + mx), the parameter b (the y value when x = 0) ranged from 46.668 in ‘Galega Vulgar’, followed by ‘Picual’ and ‘Cobrançosa’ with 50.045 and 52.085, respectively, and finally ‘Arbequina’ with 54.195 (Figure 5).

## 3. Discussion

Delineating chilling and heat accumulation needs of olive trees is one of the most challenging topics in understanding how temperatures drive the timing of bud burst and bloom in this species. Here, we investigated this question using inflorescence development and flowering data collected over three and nine years on 17 and 4 cultivars, respectively, in the Elvas region at the Olive Reference Collection of Portugal. The variability of inter-annual temperatures was the main factor explaining the cultivars’ developmental patterns, and distinct reactions to the prevailing climatic conditions were observed. Tools capable of predicting how temperature affects stages of the reproductive growth of olive trees are important for agronomic management purposes to foresee future climate impact and proactively act toward adaptation and mitigation strategies.

Elvas is a Mediterranean climate region, and so drought episodes/months are quite regular during the summer but unusual during the winter and early spring. The rainfall average has presented a high interannual variation within the Mediterranean area. In the province of Córdoba (Spain) [32] it has been observed that the number of days of rainfall in early spring has declined and recorded rainfall has dropped. In Tassaout region, Morocco, during the period 1972–2019, the lowest recorded value was 29.5 mm in 1983 and the highest was 468.8 mm in 2018 [33]. In Elvas, a similar irregular climatic behavior was observed. For example, April and May 2013 (Flw 2013) were dry months, as well as May 2014 (Flw 2014) (Figure 1) as their rainfall bars were lower than the month’s temperature (*p* < 2 × T) [36]. The worst situation occurred in 2012 (Flw 2012): the only month before the end of May that was not dry was April (Figure 1). On the other hand, in March 2013, 124.54 mm of rainfall was recorded, exceeding the month’s average for the RP (1983–2014) (Table 1). Rainfall was a very heterogeneous weather-related parameter, highlighting that, in the Elvas region, the rain episodes tended to be more concentrated and/or with very high records.

The climatic irregularity observed in Elvas throughout Flw 2012–2014 has been occurring in other Mediterranean regions [4,25,32,37]. In the province of Córdoba, since 1960, a less marked increase in air temperature during the first 3 months of the year has been observed, and there was even a drop in Tmin [32]. Similar Tmin behavior was observed in Elvas (Figure 1). In 2012, for example, the Tmins of January and February were 2.28 °C and 4.55 °C, respectively, under the Tmin average of the RP (Table 1). The sharpest increase in Córdoba’s province temperature was recorded in 2003 in late spring [32]. During the first 5 months of the year in Elvas, throughout Flw 2012–2014, the biggest increases in Tmax were observed in May of 2012 and 2014; their values were meaningful higher than the RP average, 2.32 °C and 2.25 °C greater, respectively (Table 1). In *Olea europaea* L., flowering is conditioned by climatic conditions, with temperature being the main factor [9,11], so local climatic changes are naturally impairing olive cultivars’ adjustment skills and resilience as concerns inflorescence growth and flowering.

The phenological response of the group of 17 olive cultivars during 3 consecutive years (2012–2014) demonstrated how inflorescence emergence phases can take place differently between cultivars and years (Figure 1 and Table 2). Two main features could be highlighted: (1) in general, the average amplitude between stages 51–55 decreased as a consequence of stage 51’s more delayed appearance (Table 2); (2) delay of occurrence of stage 51, observed in 2014, was intriguing as both the year’s Tmax and Tmin were very gentle and warm, without even the Tmin drop during January–February that had been observed in 2012 and 2013 (Figure 1).

Reproductive phenological response to temperature oscillations in winter and spring was clear. After the end of endodormancy of the floral buds, the time to external inflorescence development and flowering is influenced by temperature behavior [3,13]. The sigmoidal shape of the curve (Figure 2) reflects a slower response for early bud phenological stages (51–55/57), compared with the short duration of flowering phases (from stage 60). Stage 55 worked like an inflection point; during bud phenological amplitudes 51–55, a slower response was observed, but after it the duration of development phases was much shorter as the Tmax increased faster [9,11]. The fact that phenological behavior followed a sigmoidal curve in each year indicates a parallel behavior pattern in this specific site (ORCP), and the time-lag could be associated mainly with varietal characteristics (genotype).

The season Flw 2012 had the driest periods and a markedly low Tmin in January and February (Figure 1). Near March, the Tmin began to increase and overturn the inhibitory effect of cold (ecodormancy) [3] on the inflorescence buds growing. March 2013 was abnormally rainy, which had an effect on the average Tmax, in that it stayed around 15 °C during the first half of April; in the other years studied, at that time, the Tmax was already near 20 °C (Figure 1). This explains the slower progress of phenological growth stages [9] until stage 55, mainly for those cultivars that seemed to have been pushed to start the inflorescence buds growing between DOY 44 and 50; the remaining cultivars started about 10 days later (Figure 2). The effects of mild temperatures on olive flowering phenology were noticeable in a study carried out in Andalucía and Tenerife, Spain [38]. Full flowering dates of both ‘Arbequina’ and ‘Picual’ occurred much earlier in Tenerife than in Andalucía. Furthermore, flowering periods were much more prolonged in Tenerife; up to nine different phenology stages were even observed in single branches [38]. The authors suggested the lack of enough winter chilling in Tenerife caused the observed asynchronization on the flower bud burst that most likely will generate negative impacts on yield. In Elvas, the chilling requirements are fulfilled during winter; therefore, inflorescence development phases being more prolonged has not been a big concern and no particular negative effects have been associated or observed in fruit sets. Flw 2014 was the season in which inflorescence buds growth (stage 51) started later (Figure 2 and Table 1). The climate analysis of Flw 2014 (Figure 1) did not point out an obvious reason for the phenological response observed. The year 2014 was the warmest period, compared with 2013 and 2012, which, somehow, rejects the ecodormancy effect [3,13,14].

In an attempt to understand these dynamics, four cultivars (‘Arbequina’, ‘Picual’, ‘Galega Vulgar’ and ‘Cobrançosa’) were chosen and their reproductive phenology studied during a larger period (9 seasons: Flw 2012–2014 and Flw 2017–2022). Through these 9 years of collecting phenological data, the variability of manifestation of both stages 51 and stage 55 was greater among years (Figure 3a,c) than among the cultivars (Figure 3b,d). However, for stage 55 (Figure 3b) the dispersion of dates was more heterogeneous than it was for stage 51 (Figure 3d). This led us to consider a huge environmental effect related to stage 51. The variability of inter-annual temperatures was the main factor explaining the chilling and forcing periods of the ‘Picholine Marocaine’ cultivar, whereas long-term datasets were less effective in explaining this pattern [33]. If stages 51 and 55 occurrence had ups and downs during the years studied, the length of the phenological growing between stages 51–55 appeared to have two phases: a bigger one in 2012 and 2013; after that, a shorter interval stage 51–55 in 2014 and from 2017 (Figure 4a). In general, the cultivars with most and least variability were ‘Galega Vulgar’ and ‘Cobrançosa’, respectively (Figure 4b). The length of stages 51–55 was shorter in ‘Arbequina’ (Figure 4b), which means it is characterized by a faster phenological growth, not totally dependent of the end of the encodormancy. In Elvas’ climatic conditions, ‘Arbequina’ is a cultivar with precocious flowering and some authors linked that with lesser chilling requirements than, for example, ‘Picual’ [8].

The Flw 2014 was an Interesting season and the pronounced Tmin drop during November and December (Figure 2 and Table 1) suggested another explanatory hypothesis. In ‘Arbequina’, it is reported that chilling temperatures below 2.2 °C caused inflorescences development delay, compared to temperatures between 4.4 °C and 7.8 °C [31]. According to the authors, very low chilling temperatures increase the time necessary for buds’ dormancy release [31]. Indeed, the date of the manifestation of stage 51 showed a negative correlation with the Tmin of November and December, not only for ‘Arbequina’, but also for the other three cultivars (Table 3); no correlation was found with temperatures of January, February or March. This probably means that, prior to January, the four cultivars were ready to bud burst in regards to chilling accumulation and now they were leading with heat accumulation and ecodormancy. Additionally, a very good linear regression was achieved for each cultivar between the date of stage 51 and the number of days with Tmin ≤ 2.2 °C (Figure 5). The occurrence of high temperatures during dormancy causes partial deletion of the cold already accumulated [8,24], but whether very low temperatures during dormancy can have similar effects is less studied. It seems very plausible that in ‘Arbequina’, during chilling accumulation, temperatures below 2.2 °C can have a similar effect as temperatures above 20.7 °C, as described by De Melo-Abreu et al. [8] and other researchers using the same or similar methodology [2,10,17]. When air temperature is 0 °C or below, chilling units are not accumulated [8]; these authors did not consider a Tmin limit below which the accumulated cold could be deleted. Assuming the existence of a critical Tmin in ‘Arbequina’, the other olive cultivars most likely have a limit too; however, a different value is expected. According to some authors, the phases ranging from the beginning of inflorescence development (stage 51) to flowering can be simulated with high accuracy with no appreciable difference among linear and nonlinear functions [18]. In fact, we achieved very interesting results using the phenological data of four cultivars and maximum and minimum temperatures behavior since dormancy.

The lower chilling (and heat) needs of ‘Arbequina’ [8] explain its precocious flowering period in Elvas [39], so it was expected that all its intermediate stages until flowering would occur before the others. However, stage 51 occurred at the same time or even after other cultivars with higher chilling requirements (Figure 3d). This phenological growth adjustment or rhythm is reported in other regions. Thermal accumulation patterns in olive trees are strongly associated with the bioclimatic conditions in the olive-growing areas, so each orchard location showed its own particularities due to its bioclimatic peculiarities within the Mediterranean macroclimate [14,38]. In central Tunisia, it was reported that 2014 was a warmer year, and among six olive genotypes, the one that bloomed precociously was also the one with a later bud burst date (DOY 64) [40]. It seems that in ‘Arbequina’, the inflorescence’s development phases are quicker and, eventually, are implicated in the physiological mechanisms by which floral buds only overcome the state of ecodormancy when the thermal time (heat accumulation) is in an advanced state. On the other hand, there are ‘Picual’ and ‘Galega Vulgar,’ which, even with greater chilling requirements, in some years (as 2013) underwent stage 51 about 10 days prior to ‘Arbequina’ (Table 2). This “phenotypic plasticity” suggests that ‘Picual’ and ‘Galega Vulgar’ are more easily stimulated by gentle temperature conditions (end of ecodormancy) after the end of endodormancy. ‘Galega Vulgar’ and ‘Cobrançosa’ have similar climatic requirements [41], so an important aspect is that they both had to have completed stage 51 with thermal time accumulations below those of ‘Arbequina’ and ‘Picual’, as the chilling needs of Portuguese cultivars are greater [41]. ‘Arbequina’ has spread along the traditional and new olive growing areas because of its agronomical advantages (e.g., regular fruit harvests and high oil yields) [1,42]. In Italy, olive cultivars affected by “phenotypic rigidity” manifested fewest problems in terms of flower fertility [37]. These plants will be less exposed to potential climatic changes and their floral organs development and flowering periods more protected. This hypothesis can further uncover why agronomical performance of ‘Arbequina’ has been so popular among producers. However, drought episodes during late winter and spring could be very dramatic for inflorescence development and flower quality [43]. The impacts of water stress and temperature increase on olive yield has revealed critical processes differing within areas; in Seville—which has mild winters—cultivars with an earlier flowering date should be recommended, but in Granada—which has cold winters and low rainfall—the introduction of deficit irrigation strategies would be the appropriate recommendation [10].

Climate change is expected to have significant impacts on the olive oil market, where climatic conditions are already very warm and dry [8,25,40]. Drip irrigation has allowed one of the biggest limiting factors, mainly in the Alentejo, to be overcome: scarce water sources in summer; but climate change may threaten this crop through other weather parameters than extremes temperatures. Recently, a study was performed about the impact of climate change on the potential yield of olive crop over the Côa region [34]. It is a region located some miles away from Elvas, but both share some climatic traits, such as warm and dry summers and wet and cold winters. The authors reported enthusiastic news; the developed models showed an increase of potential yields and significantly improved the regional overall growth conditions [34]. Climate change could have this less negative effect, but the particularities of the region are crucial for this result. A large portion of the Côa region has shown low suitability for olive production, most likely due to low temperatures at high elevations, which will tend to improve alongside air temperature increases [34]. Presently, there is low suitability for olive groves for the rest of the olive-growing areas and as a consequence their potential yields tend to decrease. Other models with a wide area of calibration reported that an increase of both daily maximum and minimum temperatures of 2 °C and 3 °C would result in some cultivars, which require more chilling, stopping flowering [8]. Even a 1 °C mean temperature rise would lead to some years with delayed flowering dates [8]. For the same location, when stage 51 emerges, the thermal time (heat accumulation) has to be more advanced in cultivars with lower chilling requirements, compared to those with greater chilling requirements. This varietal phenotypic characteristic could be a key-factor in selecting cultivars for growing regions with, for example, high probability of frost damage during unusually late periods of freezing in the spring. 

## 4. Materials and Methods

### 4.1. Study Area

Elvas is a municipality located in Alentejo, a region in southern Portugal close to the border with Spain. The experimental field is part of the Olive Reference Collection of Portugal (ORCP), located at Olivicultura–Herdade do Reguengo, National Institute for Agrarian and Veterinarian Research, I.P. (INIAV), Pole of Elvas (latitude: 38°53′ N, longitude: 7°09′ W, 200 m a.m.s.l.). This area shows typical Mediterranean region weather. The annual mean temperature is 16.3 °C and annual rainfall 535.4 mm. The daily weather variables–maximum air temperature (Tmax; °C), minimum air temperature (Tmin; °C) and total rainfall (mm)–were obtained from a weather station located at Herdade do Reguengo, close to ORCP. A reference period (RP) for Elvas’ climate conditions was constructed based on the average weather experienced over a 30 year period (1983–2014). Because bud burst in olive is affected by previous autumn–winter temperatures, the RP began with data from October 1983.

### 4.2. Plant Material

The olive (*Olea europaea* L.) cultivars used in the present experiment were installed in the ORCP where they were watered by drip irrigation from spring to the end of summer. A group of 17 cultivars was selected, included Spanish (SP) and Portuguese (PT) genotypes from the countries’ different olive growing areas. PT: ‘Blanqueta de Elvas’, ‘Carrasquenha de Elvas’, ‘Cobrançosa’, ‘Conserva de Elvas’, ‘Cordovil de Castelo Branco’, ‘Cordovil de Elvas’, ‘Cordovil de Serpa’, ‘Galega Vulgar’, ‘Maçanilha de Tavira’, ‘Madural’, ‘Negrinha’, ‘Redondil’, ‘Verde Verdelho’, ‘Verdeal de Serpa’ and ‘Verdeal de Trás-os-Montes’; SP: ‘Arbequina’ and ‘Picual’. Phenological data were recorded from 25 year old trees planted on their own roots with a plantation compass of 5 m × 2 m. All the trees from each cultivar were obtained from one single plant.

### 4.3. Phenology Data

Reproductive phenological data of 17 olive cultivars (two trees per cultivar) was analyzed during 3 consecutive years (2012–2014). The observations were carried out every 3 days, using the procedure outlined by Fernandez-Escobar and Rallo [44]. When the beginning of flowering was approaching, the time between observations decreased. On each recording date, and for every selected tree, three observations were made: most frequent or dominant stage (X_D_), as well as less and more advanced ones (X_d_ and X_a_, respectively). A date record per tree was formed by three proportions of phenological stages (X_d_–X_D_–X_a_). Field observations were recorded using the international standardized BBCH scale for olive [45]. The onset of each phenophase of the inflorescence emergence was designated as the first Julian day (day of the year after 1 January; DOY) when that phenophase had become dominant in tree canopy. Data were collected from dormancy (stage 50) to the end of flowering (stage 69); in 2012, observations stopped on stage 55 because trees were pruning late. The following seven BBCH stages constitute the principal growth stage 5 (inflorescence emergence):

50—Inflorescence buds in leaf axils are completely closed, still in dormancy period.

51—Inflorescence buds start to swell.

53—Inflorescence buds open, starting the flower cluster development.

54—Flower cluster growing.

55—Flower cluster totally expanded. Floral buds start to open.

57—Corolla, green colored, is longer than the calyx.

59—Corolla changes color from green to white.

The onset of each phenophase of the flowering was designed according to Barranco et al. [11]. The following six BBCH stages constitute the principal growth stage 6 (flowering): 

60—First flowers open.

61—Beginning of flowering: 10% of flowers open.

65—Full flowering: at least 50% of flowers open.

67—First petals falling.

68—Majority of petals fallen or faded.

69—End of flowering, fruit set and non-fertilized ovaries fallen.

In order to simplify designations, in this paper “Flowering year 2012” (Flw 2012) is used, for example, to refer the entire path between dormancy (October 2011) and flowering in 2012. In 2017 and until 2022, the phenological observations continued, but only for four cultivars (‘Arbequina’, ‘Picual’, ‘Galega Vulgar’ and ‘Cobrançosa’). During this second phase of the study, the weather-related parameters Tmin and Tmax continued being recorded, but their behavior was not characterized as it was for the Flw 2012–2014 period. 

### 4.4. Statistical Analysis

The monthly averages of Tmax and Tmin and monthly total rainfall of the seasons Flw 2012, Flw 2013 and Flw 2014 were compared with the RP using an independent sample t-test (*p* < 0.05) (SPSS Statistics 17.0). To examine the correlation between the date of occurrence of stage 51, stage 55 or even the duration of stage 51–55, and the monthly averages of both Tmax and Tmin, we assessed the corresponding Pearson’s coefficient (*p* < 0.05) (SPSS Statistics 17.0). The strength of the correlation relationship can show six levels: very weak ]0–0.2[; weak [0.2–0.4[; moderate [0.4–0.6[; strong [0.6–0.8[; very strong [0.8–1[ and excellent (1) [46]. Linear regression trend analysis was performed for date of occurrence of stage 51 and the number of days with Tmin ≤ 2.2 °C (SPSS Statistics 17.0). In all analyses, the necessary procedures were taken into account to identify outliers and to confirm that the normality was assessed, and the variance was common and normally distributed. 

## 5. Conclusions

The interval between stages 51–55 has shortened in recent years in Elvas, compared to years 2012 and 2013. In January and February, abnormally low minimum temperatures have been recorded, which contribute to both a longer ecodormancy effect and later stage 51 manifestation by olive cultivars. Additionally, abnormally low minimum temperatures during the chilling accumulation period seemed to affect the date of the cultivars’ bud burst. Achievement of stage 51 was delayed under this condition, even when the environmental conditions were encouraging ecodormancy end. The inflorescence growth pattern, respecting temperature behavior, showed that some cultivars were more responsive to the year’s weather conditions to overcome ecodormancy, and when favorable conditions were present they progressed toward bud burst quicker than others. The increase of April and May’s maximum temperatures pushed stage 55 to earlier dates, which could represent a threat to floral organ development, flower quality, stigma pollination, fruit set and total fruit production. Olive cultivars have proven to have a high resilience to adverse climatic conditions, but future scenarios may continue challenging them. More studies linking floral bud endodormancy with specific physiological networks are necessary to determine what drives the transition from endo to ecodormancy. The next steps for this research line will be the calculation of cultivars needs regarding chilling hours and thermal time, and to identify as precisely as possible the bud burst occurrence on that timeline. Additionally, parallel studies focused on bud hormonal content and morphologic changes under microscopy would help to understand specific varietal development strategies. 

## Figures and Tables

**Figure 1 plants-12-02086-f001:**
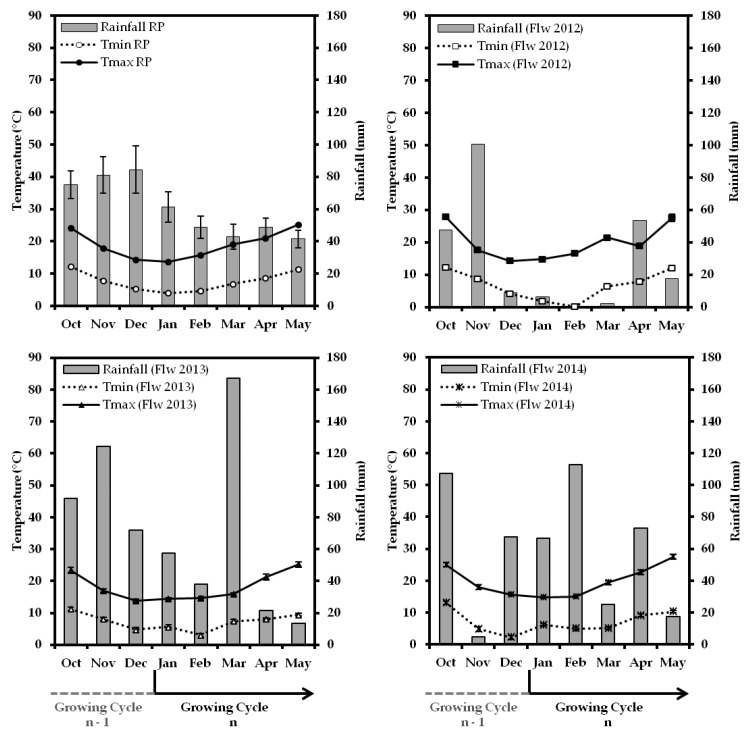
Average monthly temperatures and total rainfall recorded from October to May for the reference period (RP) 1983–2014 and for 3 consecutive years (2012, 2013 and 2014) in Elvas, Portugal. Tmax and Tmin represent average maximum and minimum temperatures, respectively. Vertical lines are standard error bars. Flw 2012/2013/2014 indicate the growing cycle (n) of flowering years of 2012/2013/2014 since previous October (n − 1).

**Figure 2 plants-12-02086-f002:**
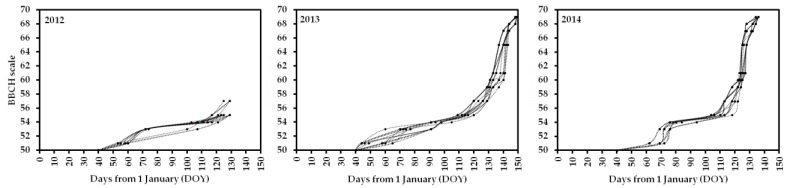
Phenological development of inflorescence growth and flowering of 17 olive cultivars grown in Elvas (Alentejo, Portugal) in 3 consecutive years (2012, 2013 and 2014). Flowering dates data not available for 2012.

**Figure 3 plants-12-02086-f003:**
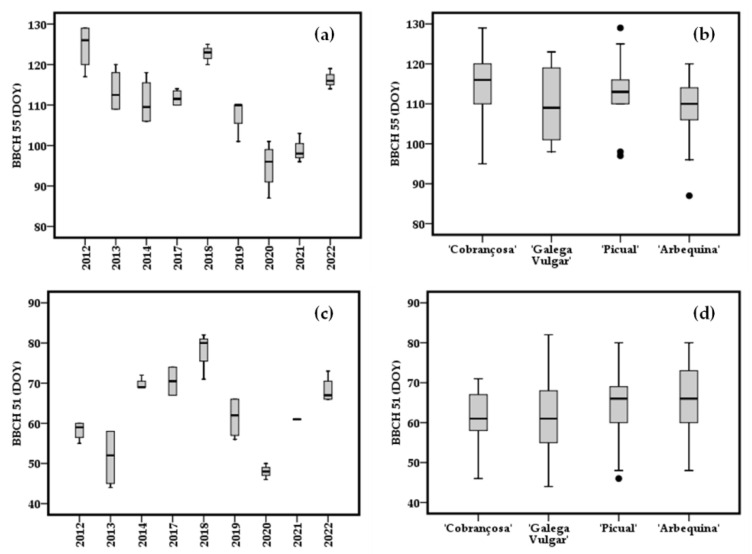
Observed distribution (box plots) of the phenological stages 51 (**c**) and 55 (**a**) with years, and with cultivars (**b**,**d**) during the same time. The boxes show the 25th and 75th percentiles, and the line inside each box represents the median. Black circumferences are outliers.

**Figure 4 plants-12-02086-f004:**
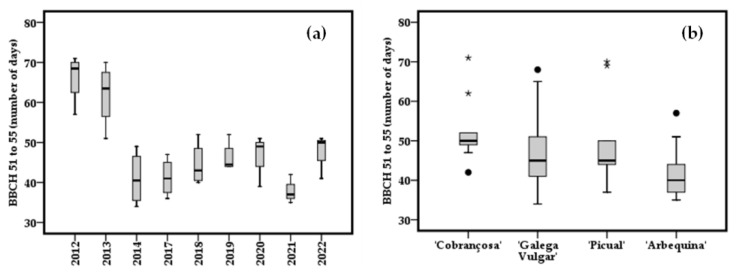
Observed distribution (box plots) of the phenological amplitude between stages 51 and 55 (**a**) with years, and with cultivars (**b**) during the same time. The boxes show the 25th and 75th percentiles, and the line inside each box represents the median. Black circumferences and * are outliers.

**Figure 5 plants-12-02086-f005:**
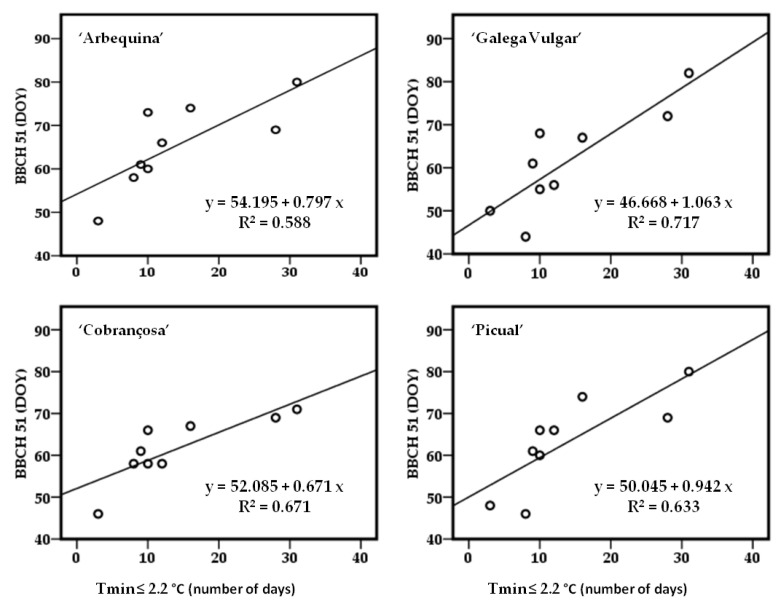
Linear regression between the occurrence of stage 51 and the number of days with minimum temperature (Tmin) equal or less than 2.2 °C during the months October–November–December previous to the flowering year. Data from nine seasons were used (Flw 2012–Flw 2014 and Flw 2017–Flw 2022).

**Table 1 plants-12-02086-t001:** Quantity of rain (mm) and Celsius degrees (°C), above or under (−) the average values of rainfall and maximum (Tmax) and minimum temperatures (Tmin) of the reference period (RP: 1983–2014), for the flowering years (Flw ^1^) of 2012, 2013 and 2014.

	October	November	December	January	February	March	April	May
Rainfall (mm)								
Flw 2012	−27.60 *	19.27 *	−75.63 *	−54.97 *	−48.62 *	−40.66 *	5.06	−23.73 *
Flw 2013	16.60	43.07 *	−12.63	−3.57	−10.62	124.54 *	−27.14 *	−27.93 *
Flw 2014	32.20 *	−76.43 *	−16.93	5.33	64.18 *	−17.46	24.26 *	−23.93 *
Tmax (°C)								
Flw 2012	3.79 *	−0.26	−0.12	1.05 *	0.80	2.31 *	−2.17 *	2.32 *
Flw 2013	−0.69	−0.91	−0.49	0.66	−1.00 *	−3.11 *	0.29	−0.08
Flw 2014	0.95	0.14	1.35 *	1.09 *	−0.73 *	0.29	1.62 *	2.25 *
Tmin (°C)								
Flw 2012	0.15	0.95	−1.27	−2.28 *	−4.55 *	−0.54	−0.84	0.62
Flw 2013	−0.85	0.27	−0.54	1.59 *	−1.58 *	0.63	−0.60	−1.92 *
Flw 2014	1.01	−2.86 *	−3.07 *	2.18 *	0.33	−1.71 *	0.52	−1.04 *

^1^ Flw 2012/2013/2014 indicate the growing cycle (n) of flowering years of 2012/2013/2014 since previous October (n − 1). * Month value is significantly different from the RP (*p* < 0.05).

**Table 2 plants-12-02086-t002:** Dates of the occurrence of stage 51 respecting a group of 17 olive cultivars and the time (number of days) between stage 51 and 55, stage 55 and beginning of flowering (60/61), and from the beginning of flowering until full flowering (65), in 3 consecutive years (2012, 2013 and 2014).

	BBCH 51 (DOY ^1^)			BBCH 51–55 (Number of Days)			BBCH 55–60/61 (Number of Days)			BBCH 60/61–65 (Number of Days)		
	2012	2013	2014	2012	2013	2014	2012	2013	2014	2012	2013	2014
Average	56	53	69	68	61	40	-	20	15	-	6	3
SE ^2^	1	2	1	2	2	1	-	1	1	-	1	0
Maximum	60	65	72	76	70	51	-	26	19	-	9	4
Minimum	53	44	62	57	49	34	-	13	7	-	2	2
‘Arbequina’	60	58	69	57	51	37	-	22	18	-	6	4
‘Cobrançosa’	58	58	69	71	62	49	-	20	7	-	2	3
‘Galega Vulgar’	55	44	72	68	65	34	-	22	17	-	6	2
‘Picual’	60	46	69	69	70	44	-	24	12	-	3	3

^1^ Day of the year (DOY) after 1 January. ^2^ Standard error. - Data not available.

**Table 3 plants-12-02086-t003:** Pearson’s correlation factor between phenological stages 51, 55 and the interval stage 51–55 with temperature (maximum and minimum) of the previous and the current growing cycle (November–December and January–April, respectively) for each cultivar.

		November		December		January		February		March		April	
		Tmax	Tmin	Tmax	Tmin	Tmax	Tmin	Tmax	Tmin	Tmax	Tmin	Tmax	Tmin
BBCH 51 (DOY)	‘Arbequina’	0.623	−0.840 **	0.285	−0.728 *	0.368	−0.215	−0.096	−0.242	−0.290	−0.402	-	-
	‘Cobrançosa’	0.590	−0.728 *	0.112	−0.733 *	0.169	0.065	−0.330	−0.094	−0.346	−0.355	-	-
	‘Galega Vulgar’	0.807 **	−0.764 *	0.489	−0.716 *	0.195	−0.093	−0.036	−0.010	−0.218	−0.421	-	-
	‘Picual’	0.694 *	−0.772 *	0.370	−0.793 *	0.230	−0.348	−0.047	−0.168	−0.025	−0.611	-	-
BBCH 55 (DOY)	‘Arbequina’	-	-	-	-	-	-	−0.256	−0.788 *	−0.376	−0.024	−0.197	−0.510
	‘Cobrançosa’	-	-	-	-	-	-	−0.524	−0.749 *	−0.450	0.177	−0.403	−0.182
	‘Galega Vulgar’	-	-	-	-	-	-	−0.083	−0.783 *	−0.499	0.366	−0.381	−0.440
	‘Picual’	-	-	-	-	-	-	−0.367	−0.865 **	−0.447	0.215	−0.433	−0.379
BBCH 51–55 (number of days)	‘Arbequina’	-	-	-	-	0.145	0.032	−0.234	−0.788 *	−0.144	0.501	−0.738 *	−0.212
	‘Cobrançosa’	-	-	-	-	0.133	0.064	−0.333	−0.810 **	−0.230	0.526	−0.739 *	−0.171
	‘Galega Vulgar’	-	-	-	-	0.156	−0.051	−0.032	−0.631	−0.190	0.722 *	−0.572	−0.222
	‘Picual’	-	-	-	-	0.104	0.266	−0.283	−0.613	−0.375	0.758 *	−0.720 *	−0.068

* Correlation is significant at the 0.05 level (2-tailed). ** Correlation is significant at the 0.01 level (2-tailed). - Data not measured.

## Data Availability

Not applicable.

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
