# Peer review of "Inflorescence Emergence and Flowering Response of Olive Cultivars Grown in Olive Reference Collection of Portugal (ORCP)"

_plants, 2023, doi:10.3390/plants12112086_

Round 1

Reviewer 1 Report

The article is interesting, but it needs some improvements:

Brief information related to the research methodology must be entered in the Abstract.

Please respect the technical editing rules of the journal. There are some spelling mistakes that need to be corrected.

The literature review section is very weak; please revise it and add new recent references.

At Results, in tables, all abbreviations must be explained in footnotes.  The figures must be revised to a better quality, the data can be grouped to reduce the number of small figures.

Discussion section should be improved by first stating the purpose of the current study and outlining the key findings. Then, these key findings should be discussed by drawing inferences or comparisons from previous study findings and highlighting implications to the olive grown. 

Reviewer 2 Report

Inflorescence emergence and flowering response of olive culti-vars grown in Olive Reference Collection of Portugal (ORCP)

            This study evaluated the reproductive phenology of 17 olive cultivars grown in Elvas (Portugal) in 3 consecutive years (2012 – 2014). Through 2017 – 2022, the phenological observations continued with 4 cultivars. The results showed that date of bud burst showed a negative correlation with minimum temperature (Tmin) of November – December, and in ‘Arbequina’ and ‘Cobrançosa’, the interval stage 51 – 55 showed a negative correlation with both Tmin of February and Tmax of April, whereas in ‘Galega Vulgar’ and ‘Picual’ there was instead a positive correlation with Tmin of March. These two seemed to be more responsive to early warm weather, whereas ‘Arbequina’ and ‘Cobrançosa’ were less sensitive. This investigation revealed that olive cultivars behaved diferently under the same environmental conditions and, in some genotypes, the ecodormancy release may be linked to endogenous factors in a stronger way. The authors concluded that Olive cultivars have proven to have a high resilience to adverse climatic conditions, but future scenarios may continue challenging them.

The MS is interesting, well written and well conducted. I appreciate the statistical work and correlation analysis. Few comments and suggestions are as follows:

-         In Abstract the authors defined the stage 51 as (Bud burst). However, they mentioned the stage 55 without definition. Abstract should be self-explanatory.

-         Line 107. To deepen this purpose, were selected 4 cultivars. Please revise.

-         Line 504. The conclusion is a little bit long. Please concentrate about your findings and recommendations.

-         References. Generally, the most of the cited references are not up to date. I recommend authors to support the discussion with recent published reports.

-         Line 555. 2008 should be Bold (2008).

-         Line 559. 2008 should be Bold (2008).

-         Line 599. 2009 should be Bold (2009).

-         Line 606. 2010 should be Bold (2010).

-         Line 608. 2008 should be Bold (2008).
